# Evaluation of Pretreatments and Extraction Conditions on the Antifungal and Antioxidant Effects of Garlic (*Allium sativum*) Peel Extracts

**DOI:** 10.3390/plants12010217

**Published:** 2023-01-03

**Authors:** David Fernando Carreón-Delgado, Itzel Yoali Hernández-Montesinos, Karla Nallely Rivera-Hernández, María del Sugeyrol Villa-Ramírez, Carlos Enrique Ochoa-Velasco, Carolina Ramírez-López

**Affiliations:** 1Instituto Politécnico Nacional, Centro de Investigación en Biotecnología Aplicada, Carretera Estatal Santa Inés Tecuexcomac-Tepetitla, km 1.5, Tepetitla de Lardizábal, Tlaxcala 90700, Mexico; 2Facultad de Ciencias Químicas, Benemérita Universidad Autónoma de Puebla, Puebla 72000, Mexico

**Keywords:** phytopathogens, antifungal, garlic peel, extraction process

## Abstract

This study aimed to evaluate the effect of pretreatments and extraction conditions on the antioxidant and antifungal characteristics of garlic peel extracts. The effect of pretreatments (fermentation and steam cooking) on the yield, antifungal (*Colletotrichum gloeosporioides* and *Botrytis cinerea*), and antioxidant (total phenolic compounds, total flavonoids, and antioxidant capacity) properties of garlic peel extracts were evaluated. A selected pretreatment was applied to evaluate the extraction conditions (solvent, solvent concentration, temperature, and time) on the antifungal activity of garlic peel extracts. At last, garlic peel extracts obtained under specific conditions was applied to papaya and strawberry fruits as preventive and curative treatments against *Colletotrichum gloeosporioides* and *Botrytis cinerea*, respectively. Steam cooking pretreatment significantly increased the antifungal and antioxidant capacities of garlic peel extracts compared to the fermentation process. Garlic peel extracts obtained with methanol (60%) for 18 h (25 °C) showed the highest antifungal activity against both microorganisms assessed (57.57% and 75.76% for *B. cinerea* and *C. gloeosporioides*, respectively,) on in vitro assays. Moreover, in vivo results indicated that preventive treatment significantly reduced rot disease in papaya (88.95%) and strawberry (54.13%) fruits. Although more studies about the antifungal mechanisms of garlic peel extracts are needed, these results indicated that garlic peel extracts could be used as an antifungal agent.

## 1. Introduction

Plants and their components are one of the principal sources of bioactive compounds with potential applications in food, pharmaceutical, and biotechnological industries. Bioactive compounds such as phenolic acids, flavonoids, terpenoids, carotenoids, and some vitamins are synthesized from the secondary metabolism of plants when they are stressed by biotic or abiotic factors [1]. Among different properties attributed to the bioactive compounds, “functional” properties such as antioxidant, antidiabetic, anticancer, antimicrobial, and anti-hyperglycemia are the most important. Moreover, the Food and Drug Administration (FDA) classified the bioactive compounds of some herbal plants and their extract as generally recognized as safe (GRAS) [2]. The extraction process is the first step in procuring, isolating, and characterizing bioactive compounds from plant sources. This process affects the quantitative and qualitative characteristics of the extracts, so its evaluation is of paramount importance [3]. Some studies have indicated that the treatments or pretreatments applied to the sample, the operating parameters (solvents, temperatures, time, etc.), and the extraction processes (traditional or novel technologies) are the main factors for the extraction of bioactive compounds; and their selection generally depends on the characteristics of the sample [3,4].

Garlic (*Allium sativum*) is a plant belonging to the Liliaceae family; its consumption is due to its culinary value and medicinal properties [5]. Garlic consumption is generally associated with several human health benefits such as cardioprotective, antimicrobial, antioxidant, anti-inflammatory, antiviral, and anti-aging, among other benefits, associated with its biologically active compounds, such as allyl sulfur compounds, terpenoids, and phenolic compounds [6]. The generation of garlic by-products due to industrial processing poses environmental and economic difficulties. Peel is one of the main problems since it represents approximately 24% of the total weight. It is generally discarded and has been seldomly studied so far [7]. This by-product has been used to obtain extracts and evaluate their antimicrobial [8,9] and antioxidant capacities [10,11], or to improve the antimicrobial activity of edible films and coatings [7,12,13,14]. However, to the best of the author’s knowledge, garlic peel extracts against fungal phytopathogens, such as *Colletotrichum gloeosporioides* and *Botrytis cinerea* in in vivo and in vitro assays, have not been evaluated yet. Therefore, this study aimed to evaluate the effect of pretreatments and extraction conditions on the antioxidant and antifungal activities of garlic peel extracts. To fully achieve this purpose, two topics were covered: (i) evaluate the effect of pretreatments applied on the antioxidant and antifungal activities of garlic peel extracts against *Colletotrichum gloeosporioides* and *Botrytis cinerea* on in vitro assays, and (ii) evaluate the effect of different extraction conditions on the antifungal activity of garlic peel extracts against fungal strains in both in vitro and in vivo assays in papaya and strawberry fruits.

## 2. Materials and Methods

### 2.1. Garlic Peels

Garlic peels were obtained as a by-product from a local producer in San Matias Tlalancaleca, Puebla, Mexico. Peels were washed with distilled water to withdraw any undesirable particles. Samples were oven-dried at 40 °C for 24 h. Dried peels were ground, sieved (2 mm), and stored in glass containers in a dark environment for further use.

### 2.2. Reagents and Solvents

All chemical reagents and solvents used in this study were obtained from Sigma-Aldrich, Inc. (Toluca, Mexico) and J.T. Baker (Mexico City, Mexico), respectively. Broths and agars were obtained from BD Bioxon (Mexico City, Mexico).

### 2.3. Microorganisms

*Lactobacillus plantarum*, *Colletotrichum gloeosporioides*, and *Botrytis cinerea* were obtained from the microbiological collection of the Centro de Investigación en Biotecnología Aplicada, Tlaxcala, México (CIBA-IPN). Microorganisms were stored in a deep freeze (−70 °C) until reactivation. *L. plantarum* was reactivated in MRS broth at 37 ± 2 °C and used for the fermentation process after 24 h. On the other hand, fungi were reactivated in PDA agar and growth at 25 ± 2 °C. Fungal strains were maintained at 4 °C and periodically cultivated (14 days) by punction in a new PDA agar.

### 2.4. Selection of Garlic Pretreatments

The antifungal and antioxidant activities of garlic peel extracts were studied in two parts. The first part consisted of the pretreatment selection of the sample and the second part involved the evaluation of extraction conditions.

Part one. Garlic peel was pretreated by fermentation or heat (steam cooking). The fermentation process was conducted based on the methodology proposed by Kim et al. [15] with slight modifications. Briefly, 60 g of sterilized garlic peels and 1.2 g of glucose were mixed with 120 mL of the Mandels and Weber mineral solution. Then, the mixture was inoculated with *L. plantarum* and fermented for 16 days at 30 °C. Finally, fermented garlic peels were stored at 4 °C until their use for the extraction procedure. The steam cooking process was conducted based on the methodology proposed by Min et al. [16] with some modifications. Sixty g of garlic peels were placed in a bottle resistant to high temperature and pressure for a wet thermal treatment (100 °C for 15 min). Then, the peels were toasted at 70 °C for 10 min. Finally, samples were stored at 4 °C until their use for the extraction process. Untreated garlic peel was used as a control. The extraction process was conducted by placing 1 g of untreated (control) or pretreated garlic peels with 25 mL of solvent (ethanol or methanol, 60% *v*/*v*) under agitation at 120 RPM at 50 °C for 18 h. Extracts were filtered through the Whatman number 4 paper. Samples were dried using a rotary evaporator and resuspended with distilled water (1% tween 80) until 10 mg/mL of concentration. The yield was calculated as the ratio between the sample placed in the extraction process and the dry extract obtained. The in vitro antifungal and antioxidant activities were evaluated to determine pretreatment to study the extraction conditions. 

Part two. The extraction conditions were evaluated following a 2^4^-factorial design with solvent (ethanol and methanol), concentration (60% and 80%), temperature (25 and 50 °C), and time (1 and 18 h) as experimental factors (Table 1). The optimal extraction condition was obtained by the numerical optimization tool of Design Expert Program 6.0.6 (StatEase Inc., Minneapolis, MN, USA). The extracts were handled as mentioned in the previous section.

### 2.5. Antifungal Activity

Antifungal activity was evaluated through in vitro and in vivo assays. The in vitro effect of garlic peel extract was assessed by measuring the inhibition percentage of radial mycelial growth of *C. gloeosporioides* and *B. cinerea*. Poisoned agar (1 mg of garlic extract/mL of agar) was used as the methodology to evaluate the fungal growth. The spore suspension was obtained by pouring 9 mL of sterile physiological water on the PDA agar plate surface, followed by a gentle scraping with a sterile rake to recover the maximum quantity of spores until reaching 1 × 10^6^ spores/mL, approximately [17]. Then, ten µL of fungal spores were placed in the center of a Petri dish, and their growth was measured after 7 days of storage at 25 ± 2 °C. The diameter was measured using a digital vernier, and the percentage of inhibition was evaluated according to the following equation:(1)$$\%\;Inhibition=1-\frac{x}{y}\times 100$$
where *y* and *x* are the radial growth in control and poisoned agar Petri dishes, respectively.

The in vivo assays were evaluated in papaya (*Carica papaya*) and strawberry (*Fragaria* x *ananassa*) fruits by inoculating *C. gloeosporioides* and *B. cinerea*, respectively. Fruits were selected free from physical and microbiological appearance damages from a local supermarket in Puebla, Puebla, Mexico. Fruits were washed and disinfected in a sodium hypochlorite solution (2%) for 5 min. Then, the fruits were gently dried with absorbent paper. 8Both fruits were divided into three batches. Batch 1 consisted of preventive treatment; 2 h before inoculation, fruits were three-times sprayed with 30 mL of selected garlic peel extract (10 µg/mL). Batch 2 was a curative treatment; 6 h after inoculation, fruits were three-times sprayed with 30 mL of selected garlic peel extract (10 µg/mL) on the top of the inoculated zone. Batch 3 consisted of the control group. The inoculation process was carried out by placing 10 µL of the spore suspension through one (strawberry) or three (papaya) incisions made with a sterilized stainless-steel knife. The infection process was evaluated after 7 days of storage at 25 ± 2 °C.

#### Disease Incidence

In papaya, the disease incidence was evaluated as the surface infected area using ImageJ software. The infected and total area was evaluated by analyzing color thresholds in the images. The infection severity was calculated as the ratio between the infected and the total areas [18]. On the other hand, in strawberries, disease incidence was estimated according to the 0–4 scale percentage infected where 0 = (0%), 1 = (1–25%), 2 = (26–50%), 3 = (51–75%), and 4 = (76–100%). The disease index was calculated as follows:(2)$$DI=\frac{n_{1}+n_{2}+n_{3}+n_{4}}{n}\pi r^{2}$$
where *DI* is the disease index, and *n*_1_, *n*_2_, *n*_3_… represent the number of samples cataloged according to the disease incidence (0 to 4) in each treatment.

### 2.6. Total Phenolic Compounds

Total phenolic compounds were evaluated following the methodology reported by Hernández-Carranza et al. [19]. Briefly, 1 mL of garlic peel extract was mixed with 1 mL of Folin–Ciocalteu reagent (0.1 N). The mixture was left to stand for 3 min, and then 1 mL of Na_2_CO_3_ (0.05% *w*/*v*) solution was added. The mixture was incubated for 1 h at room temperature in a dark environment. The absorbance was read at 765 nm using a Jenway UV–Vis spectrophotometer (model 6405, Staffordshire, UK). Results were expressed as milligrams of gallic acid equivalents (GAE) per g dry sample (dm) using a standard curve of gallic acid.

### 2.7. Total Flavonoids

Total flavonoids (TF) were analyzed following the methodology proposed by Hernández-Carranza et al. [19]. Then, 1 mL of garlic peel extract was mixed with 1 mL of NaNO_2_ (1.5% *w*/*v*), and the mixture was left to stand for 5 min. Then, 1 mL of AlCl_3_ (3% *w*/*v*) was added and mixed for 1 min; afterward, 0.5 mL of NaOH (1 N) was added. The mixture was incubated for 1 min, and the absorbance was read at 490 using a UV–Vis spectrophotometer. Results were expressed as mg of quercetin per g of dm using a standard curve of quercetin.

### 2.8. Antioxidant Capacity

Antioxidant capacity was evaluated by the inhibition of the 2,2-diphenyl-1-picrilhidrazil (DPPH) radical [19]. One mL of extract was mixed with 1 mL of DPPH solution (0.004%) and allowed to stand for 30 min at room temperature in the dark. Antioxidant capacity was measured at 517 nm using a UV–Vis spectrophotometer. Results were expressed as milligrams of Trolox per g of dm using a Trolox standard curve.

### 2.9. Statistical Analysis

A statistical analysis (α = 0.05) was conducted by analysis of variance (ANOVA) using Tukey’s test for pairwise comparison in Minitab 15 software (Minitab Inc. State College, Pennsylvania, PA, USA).

## 3. Results and Discussion

### 3.1. Pretreatment Effect on Antifungal and Antioxidant Capacity of Garlic Peel Extracts

The effect of pretreatment on the yield and antifungal activity of garlic peel extracts is shown in Table 2. As is observed, both treatments increased the yield of the extraction process (3.04–5.86 g extract/g dm). However, the fermentation process and ethanol as solvent showed the highest yield (*p* < 0.05) attributed to the increase in biomass and enzyme production of the solid-state fermentation [20]. On the other hand, all extracts presented antifungal effects against *C. gloeosporioides* (31.53–72.33%) and *B. cinerea* (11.61–63.91%), possibly attributed to the organosulphur compounds such as allicin, a thiosulfinate with two allyl groups as carbon chains [7]. In this aspect, allicin reacts with low-molecular-weight cellular thiols such as glutathione, shifting its state to a more oxidized condition, which causes cell apoptosis [21]. Interestingly, the fermentation process reduced the antimicrobial activity of garlic peels, whereas both untreated and cooked garlic peels showed higher antimicrobial activity against *C. gloeosporioides*. Moreover, untreated garlic peel extracts showed high antifungal activity against *B. cinerea*, which may be explained due to (i) allyl compounds presented in garlic peels being very unstable, producing many allyls and methyl sulfide derivatives, such as diallyltrisulfide (DATS) and diallyltetrasiulfide (DATTS), which have less antimicrobial activity [22,23], and (ii) *B. cinerea* may present point mutations in its genes that encode the fungicide target protein, reducing antifungal effects [24].

Though garlic peel is considered a waste, it can be a good source of bioactive compounds such as caffeic, ρ-coumaric, ferulic, and di-ferulic acids [25]. Table 3 presents the total flavonoids, total phenolic compounds, and antioxidant capacity of garlic peel extracts. As is observed in untreated garlic extract (control), the concentration of total flavonoids, total phenolic compounds, and antioxidant capacity were between 9.25 and 10.25 mg quercetin/g, 28.50–32.47 mg GAE/g, and 86.34–96.16 µmol Trolox/g, respectively. Although reports on the bioactive compounds and antioxidant capacity of garlic peels are scarce, some studies have indicated that total flavonoids were higher than the values reported for garlic bulbs (0.6–75 mg quercetin/g) obtained from different regions of Morocco [26]. The values of total phenolic compounds are very similar to the data obtained for husks (28.35 ± 0.07 mg GAE/g dm) and different varieties of garlic bulbs (17.16–42.53 mg GAE/g) [27,28]. The antioxidant capacity of garlic extract was higher than the results reported by Lu et al. [29] (6.76–10.21 µmol Trolox/g FW), highlighting that their values are on a fresh basis (moisture was not provided by the authors). As mentioned by several authors, the bioactive compounds and antioxidant capacity of fruit and vegetables are affected by different factors; among them, the variety, pre- and post-harvest management, and extraction condition and quantification are the most important [3]. On the other hand, both pretreatments significantly increased the bioactive compounds and antioxidant capacity of the extracts. It is noteworthy that steam cooking pretreatment and methanol as solvent displayed a higher antioxidant capacity (*p* < 0.05), showing an increase in the range of 63.9–74.6, 48.8–84.9, and 25.5–55.7% in total flavonoids, total phenolic compounds, and antioxidant capacity, respectively. Probably, the higher yield obtained in pretreated samples was related to the extraction of the antioxidant compounds [20]. In this sense, Ilić et al. [30] indicated that thermal treatment with methanol did not affect the antioxidant capacity of allicin and its derivatives. Therefore, it is possible to infer that other compound such as lignin (a component of the plant cell wall), might be decomposed into phenolic compounds, increasing the antioxidant capacity of garlic peel extracts [28]. Based on the above, steam cooking pretreatment was used to assess the extraction conditions of garlic peel and its application against *C. gloeosporioides* and *B. cinerea* in in vitro and in vivo assays.

### 3.2. Evaluation of Extraction Conditions on In Vitro and In Vivo Antifungal Activities of Garlic Peel Extracts

The effect of solvent, concentration, temperature, and extraction time on the inhibition of *C. gloeosporioides* and *B. cinerea* is presented in Table 4. The garlic peel extracts showed inhibition percentages in the range of 53.81 to 75.76 and 36.05 to 57.57 for *C. gloeosporioides* and *B. cinerea*, respectively. The in vitro results indicated that *B. cinerea* is more resistant than *C. gloeosporioides* to the garlic peel extracts. As far as the authors know, this is the first study about garlic peel extracts against *C. gloeosporioides* and *B. cinerea.* However, garlic clove extract was assessed against *C. gloeosporioides,* and the results indicated that at the same concentration (1000 ppm), the mycelial inhibition was 33.78%, whereas to reach a similar inhibition to this study, 5000–10,000 ppm is required [31]. Moreover, in a recent study, garlic clove extract needed higher concentrations (30 g/L–80 g/L) to reach an inhibition percentage of 76.19–82.94% after 8 days of radial growth of *C. gloeosporioides* [32]. In the latter study, the authors did not provide detailed information for an adequate calculation of the concentration used. On the other hand, the ethanolic of garlic clove extract (20–40% *w*/*v*) attained similar values of inhibition (42.17–69.20%) for *B. cinerea* to those obtained in this study [33].

According to the optimization procedure, extraction at 25 °C for 18 h using aqueous methanol (60% *v*/*v*) showed higher antifungal inhibition. Therefore, this condition was selected to obtain the extract further used for evaluating its effect against *C. gloeosporioides* and *B. cinerea* on in vitro (Figure 1) and in vivo (Figure 2) assays. After 7 days of inoculation, the inhibition percentages for *C. gloeosporioides* (papaya) and *B. cinerea* (strawberry) were 11.05–49.33% and 45.87–79.69, respectively. It is noteworthy that preventive treatment significantly reduced (*p* < 0.05) the incidence of the disease’s appearance in both fruits after 7 days of inoculation (Figure 2A). Interestingly, results showed a contrary effect to the results obtained in the in vitro assays, showing a higher inhibition of the extract against *B. cinerea* in strawberry fruits, highlighting the importance of conducting in vivo assays. Daniel et al. [34] applied garlic extracts to delay *B. cinerea* growth in apple fruits. Their results indicated that curative treatment showed a better response than the preventive treatment when garlic extracts were used at 40–50%, showing a similar inhibition of the lesion diameter than was obtained in this study. Similarly, garlic extract reduced mango rot caused by *Lasiodiplodia theobromae* in the range of 55.50% to 73.46% when fruit was soaked in garlic extract (100%) for 1 or 2 h [35].

## 4. Conclusions

Results obtained in this study indicated that despite a slight reduction in antifungal activity (*B. cinerea*) of steam-cooked garlic peel extracts being observed, this pretreatment increased the bioactive compounds and antioxidant capacity of garlic peel extracts. On the other hand, selected garlic peel extracts (methanol 60%, 25 °C for 18 h) showed acceptable inhibition percentages for in vitro and in vivo assays against *C. gloeosporioides* and *B. cinerea*, respectively. Moreover, the application of selected garlic extracts as a preventive treatment presented a significant reduction (*p* < 0.05) in fungal growth after 7 days of inoculation. Although further studies are necessary to clarify the effect of garlic peel extracts as antioxidant and antimicrobial agents, this study indicated that this by-product could be used as an antifungal agent against two phytopathogens in fruits.

## Figures and Tables

**Figure 1 plants-12-00217-f001:**
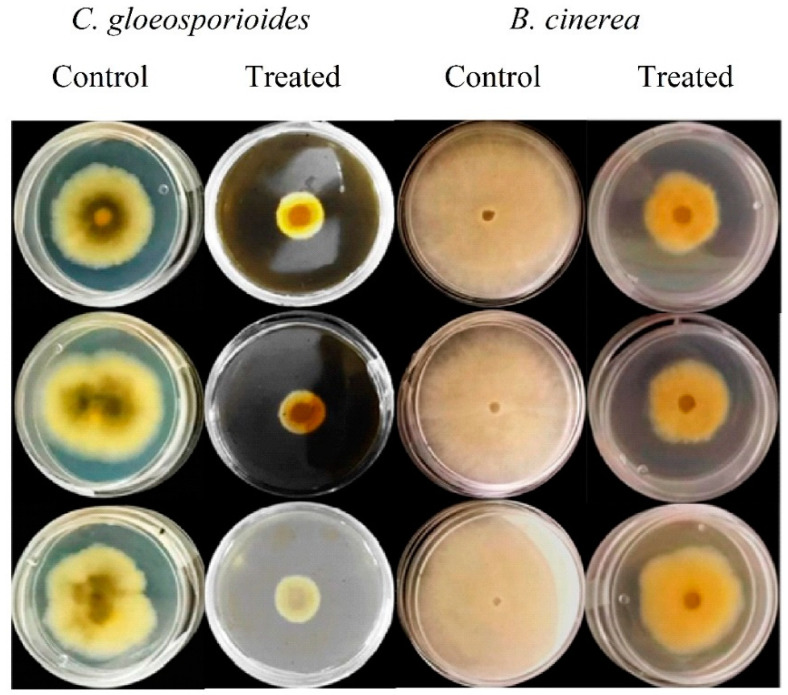
In vitro mycelial growth inhibition of pretreated garlic peel extracts against *C. gloeosporioides* and *B. cinerea* after 7 days of storage.

**Figure 2 plants-12-00217-f002:**
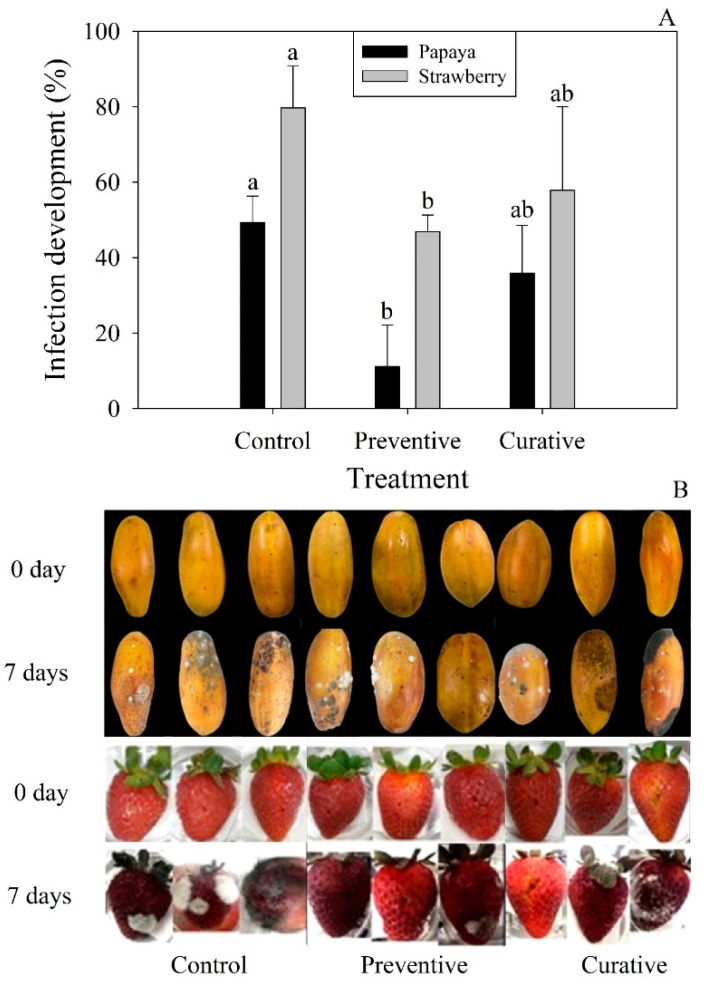
In vivo antifungal activity of garlic peel extracts against *C. gloeosporioides* and *B. cinerea* inoculated on papaya and strawberry after 7 days of storage (25 ± 2 °C). (**A**) Percentage of infection. (**B**) Fungi development. Different letters (a, b) indicate significant statistical differences (*p* < 0.05).

**Table 1 plants-12-00217-t001:** Factorial design to select the extraction conditions of pretreated garlic peels.

Factor	Level	
Solvent	Ethanol	Methanol
Solvent concentration (%)	60	80
Temperature (°C)	25	50
Time (h)	1	18

**Table 2 plants-12-00217-t002:** Effect of pretreatment on the yield and antifungal activity of garlic peel extracts ^†^.

	Yield (%)		*C. gloeosporioides* (%)		*B. cinerea* (%)	
Treatment	Ethanol	Methanol	Ethanol	Methanol	Ethanol	Methanol
Control	5.24 ± 1.90 ^a,y^	4.84 ± 0.35 ^a,x^	65.90 ± 0.28 ^b,x^	70.49 ± 0.82 ^a,x^	63.09 ± 4.28 ^a,x^	63.91 ± 4.23 ^a,x^
Fermented	11.10 ± 1.05 ^a,x^	7.88 ± 0.68 ^b,x^	31.53 ± 2.54 ^a,y^	36.03 ± 1.04 ^a,y^	11.61 ± 1.17 ^b,z^	36.28 ± 0.31 ^a,y^
Steam cooked	9.68 ± 2.76 ^a,xy^	8.16 ± 3.18 ^a,x^	67.71 ± 0.79 ^b,x^	72.33 ± 1.12 ^a,x^	40.60 ± 5.40 ^a,y^	41.00 ± 6.60 ^a,y^

^†^ Average (*n* = 3) ± standard deviation. Different letters (a and b) within the same line for the same characteristics are statistically different (*p* < 0.05). Different letters (x, y, z) within the same column are statistically different (*p* < 0.05).

**Table 3 plants-12-00217-t003:** Effect of pretreatment on the antioxidant compounds and capacity of garlic peel extracts ^†^.

	Total Flavonoids (mg Quercetin/g)		Total Phenolic Compounds (mg GAE/g)		Antioxidant Capacity (µmol Trolox/g)	
Treatment	Ethanol	Methanol	Ethanol	Methanol	Ethanol	Methanol
Control	9.25 ± 0.18 ^b,z^	10.25 ± 0.11 ^a,z^	28.52 ± 0.19 ^b,z^	32.47 ± 0.38 ^a,z^	86.34 ± 0.84 ^b,z^	96.16 ± 0.42 ^a,z^
Fermented	12.39 ± 0.18 ^b,y^	14.03 ± 0.30 ^a,y^	36.24 ± 0.46 ^b,y^	40.37 ± 0.79 ^a,y^	101.36 ± 0.37 ^b,y^	119.48 ± 0.50 ^a,y^
Steam cooked	16.44 ± 0.11 ^b,x^	17.90 ± 0.13 ^a,x^	38.61 ± 0.28 ^b,x^	60.06 ± 0.18 ^a,x^	121.59 ± 1.54 ^b,x^	149.75 ± 1.69 ^a,x^

^†^ Average (*n* = 3) ± standard deviation. Different letters (a and b) within the same line for the same characteristics are statistically different (*p* < 0.05). Different letters (x, y, z) within the same column are statistically different (*p* < 0.05).

**Table 4 plants-12-00217-t004:** Effect of extraction conditions on the yield and antifungal activity of garlic peel extracts ^†^.

Run	Solvent	Concentration (%)	Temperature (°C)	Time (h)	Yield	*C. gloeosporioides*	*B. cinerea*
1	M	60	25	1	9.39 ± 0.93 ^ab^	65.22 ± 0.84 ^c^	36.05 ± 2.34 ^d^
2	E	60	25	1	7.06 ± 0.77 ^ab^	72.10 ± 0.72 ^a^	55.73 ± 0.38 ^ab^
3	M	80	25	1	6.25 ± 1.44 ^ab^	53.81 ± 0.80 ^e^	36.22 ± 1.60 ^d^
4	E	80	25	1	5.75 ± 1.63 ^b^	69.55 ± 0.22 ^b^	49.04 ± 0.80 ^abc^
5	M	60	50	1	11.34 ± 1.06 ^a^	66.67 ± 3.58 ^bc^	38.52 ± 2.87 ^cd^
6	E	60	50	1	6.08 ± 0.57 ^ab^	74.49 ± 1.03 ^ab^	54.23 ± 2.97 ^ab^
7	M	80	50	1	6.87 ± 1.19 ^ab^	70.13 ± 0.88 ^b^	43.87 ± 5.35 ^bcd^
8	E	80	50	1	7.63 ± 2.17 ^ab^	62.93 ± 0.27 ^cd^	40.50 ± 3.84 ^cd^
9	M	60	25	18	7.30 ± 0.39 ^ab^	75.76 ± 1.12 ^a^	57.57 ± 5.36 ^a^
10	E	60	25	18	4.96 ± 0.72 ^b^	74.06 ± 3.10 ^ab^	54.62 ± 0.78 ^a^
11	M	80	25	18	7.43 ± 2.48 ^ab^	69.40 ± 3.24 ^b^	39.92 ± 2.32 ^cd^
12	E	80	25	18	9.76 ± 2.02 ^ab^	71.98 ± 0.76 ^ab^	38.34 ± 1.08 ^cd^
13	M	60	50	18	8.16 ± 3.18 ^ab^	72.33 ± 1.12 ^ab^	41.00 ± 6.60 ^cd^
14	E	60	50	18	9.68 ± 2.76 ^ab^	67.71 ± 0.79 ^bc^	40.60 ± 5.40 ^cd^
15	M	80	50	18	7.07 ± 2.33 ^ab^	67.32 ± 2.78 ^bc^	53.05 ± 1.35 ^ab^
16	E	80	50	18	8.00 ± 2.04 ^ab^	59.67 ± 1.34 ^d^	40.97 ± 4.01 ^cd^

^†^ Average (*n* = 3) ± standard deviation. Different letters (a, b, c, d, e) within the same column are statistically different (*p* < 0.05).

## Data Availability

Data recorded in the current study are available in all Tables and Figures of the manuscript.

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
