# Peer review of "Evaluation of Pretreatments and Extraction Conditions on the Antifungal and Antioxidant Effects of Garlic (Allium sativum) Peel Extracts"

_plants, 2023, doi:10.3390/plants12010217_

Round 1
Reviewer 1 Report
The present article "Evaluation of pretreatments and extraction conditions on the antifungal and antioxidant effect of garlic peel (Allium sativum) extract and its potential uses as a postharvest agent" deals an interesting theme.
Although the antimicrobial activity of the garlic has been well reported in the previous literature. In this article authors have taken their peel for the study. Author should characterize the compounds present in the peel, and also make a comparative study with garlic. that will provide a details and broad overview of this study
Author Response
Thank you very much for your accurate comments to improve the quality of the manuscript. We are doing our best to improve it. The required information is written in blue in the current version of the manuscript.
The reviewer is right; the garlic peel extracts must be deeply studied in their composition due to contain several bioactive compounds. Unfortunately, at this moment, we don’t have this recommended characterization, and we will obtain it in a few months (Some instruments and equipment of our laboratories are under review). However, we added some information (Table 3) and discussion about the characterization (total flavonoids, total phenolic compounds, and antioxidant capacity), which were not discussed in the first version of the manuscript (please see lines 192-221 in the new version).
New information added.
“Despite garlic peel is considered a waste, it can be a good source of bioactive compounds such as caffeic, ρ-coumaric, ferulic, and di-ferulic acids [23]. Table 3 presents the total flavonoids, total phenolic compounds, and antioxidant capacity of garlic peel extracts. As it is observed in untreated garlic extract (control), the concentration of total flavonoids, total phenolic compounds, and antioxidant capacity were between 9.25 to 10.25 mg quercetin/g, 28.50-32.47 mg GAE/g, and 86.34-96.16 µmol Trolox/g, respectively. Although reports about antioxidant compounds and the capacity of garlic peel are scarce, some studies have indicated that total flavonoids were higher than the values reported for garlic bulbs (0.6-75 mg quercetin/g) obtained from different regions of Moroccan [24]. The values of total phenolic compounds are very similar to the data obtained for husk (28.35 ± 0.07 mg GAE/g dm) and different varieties of garlic bulb (17.16-42.53 mg GAE/g) [25,26]. The antioxidant capacity of garlic extract was higher than the results reported by Lu et al. [27] (6.76-10.21 µmol Trolox/g FW), highlighting that their values are on a fresh basis (moisture was not provided by the authors). As mentioned by several authors, bioactive compounds and antioxidant capacity of fruit and vegetables are affected by different factors; among them, the variety, pre-and post-harvest management, and extraction condition and quantification are the most important [3]. On the other hand, both pretreatments significantly increased the bioactive compounds and antioxidant capacity of the extracts. It is noteworthy that cooking pretreatment and methanol as solvent displayed a higher antioxidant capacity (p < 0.05), showing an increase in the range of 63.9-74.6, 48.8-84.9, and 25.5-55.7% in total flavonoids, total phenolic compounds, and antioxidant capacity, respectively. Probably, the higher yield presented in pretreated samples was related to the extraction of the antioxidant compounds [18]. In this sense, Ilić et al. [28] indicated that methanol heating treatment did not affect the antioxidant capacity of allicin and its derivatives. Therefore, it is possible to infer that other compounds like lignin (a component of the plant cell wall) might be decomposed into phenolic compounds, increasing the antioxidant capacity of garlic peel extracts [26]. Based on the above, cooking pretreatment was used for selecting extraction conditions of garlic peel and its application against C. gloeosporioides and B. cinerea in in vitro and in vivo assays.”
(24) Ourouadi, S. ; Hasib, A.; Moumene, H.; El Khiraoui, A.; Ouatmane, A.; Boulli, A.A. Bioactive Constituents and Antioxidant Activity of Moroccan garlic (Allium sativum L.). Journal of Natural Sciences Research. 2016, 6, 38-43.
(25) Chen, S.; Shen, X.; Cheng, S.; Li, P.; Du, J.; Chang, Y.; Meng, H. Evaluation of Garlic Cultivars for Polyphenolic content, and Antioxidant Properties. 2013, Plos One, 8, e79730. https://doi:10.1371/journal.pone.0079730
(26) Lu, X.; Ross, C.; Powers, J.R.; Aston, D.E.; Rasco, B.A. Determination of Total Phenolic Content and Antioxidant Activity of Garlic (Allium sativum) and Elephant Garlic (Allium ampeloprasum) by Attenuated Total Reflectance-Fourier Transformed Infrared Spectroscopy. Journal of Agricultural and Food Chemistry. 2011, 59, 5215-5221. https://doi.org/10.1021/jf201254f |

Reviewer 2 Report
Major revision:
Where is the results of the antioxidants tests ? As mentioned in the text (sections 2.6; 2.7; 2.8), total phenolic compounds, total flavonoids, antioxidant capacity carried out but the results aren't present in the manuscript.
Please provide missing data and complete the manuscript, and then resubmit it for reviewing.
Author Response
Thank you very much for your accurate comments to improve the quality of the manuscript. We are doing our best to improve it. The required information and the revision of English language and style are written in blue in the current version of the manuscript.
Results of total flavonoids, total phenolic compounds and antioxidant capacity are presented in Table 2 and they were discussed in the lines 193-208 of the past version of the manuscript. To avoid confusion, more information and discussion were added to the current version of the manuscript (Line 192-221) and Table 2 was divided into two Tables (Table 2 and 3).
New information added.
“Despite garlic peel is considered a waste, it can be a good source of bioactive compounds such as caffeic, ρ-coumaric, ferulic, and di-ferulic acids [23]. Table 3 presents the total flavonoids, total phenolic compounds, and antioxidant capacity of garlic peel extracts. As it is observed in untreated garlic extract (control), the concentration of total flavonoids, total phenolic compounds, and antioxidant capacity were between 9.25 to 10.25 mg quercetin/g, 28.50-32.47 mg GAE/g, and 86.34-96.16 µmol Trolox/g, respectively. Although reports about antioxidant compounds and the capacity of garlic peel are scarce, some studies have indicated that total flavonoids were higher than the values reported for garlic bulbs (0.6-75 mg quercetin/g) obtained from different regions of Moroccan [24]. The values of total phenolic compounds are very similar to the data obtained for husk (28.35 ± 0.07 mg GAE/g dm) and different varieties of garlic bulb (17.16-42.53 mg GAE/g) [25,26]. The antioxidant capacity of garlic extract was higher than the results reported by Lu et al. [27] (6.76-10.21 µmol Trolox/g FW), highlighting that their values are on a fresh basis (moisture was not provided by the authors). As mentioned by several authors, bioactive compounds and antioxidant capacity of fruit and vegetables are affected by different factors; among them, the variety, pre-and post-harvest management, and extraction condition and quantification are the most important [3]. On the other hand, both pretreatments significantly increased the bioactive compounds and antioxidant capacity of the extracts. It is noteworthy that cooking pretreatment and methanol as solvent displayed a higher antioxidant capacity (p < 0.05), showing an increase in the range of 63.9-74.6, 48.8-84.9, and 25.5-55.7% in total flavonoids, total phenolic compounds, and antioxidant capacity, respectively. Probably, the higher yield presented in pretreated samples was related to the extraction of the antioxidant compounds [18]. In this sense, Ilić et al. [28] indicated that methanol heating treatment did not affect the antioxidant capacity of allicin and its derivatives. Therefore, it is possible to infer that other compounds like lignin (a component of the plant cell wall) might be decomposed into phenolic compounds, increasing the antioxidant capacity of garlic peel extracts [26]. Based on the above, cooking pretreatment was used for selecting extraction conditions of garlic peel and its application against C. gloeosporioides and B. cinerea in in vitro and in vivo assays..”
(24) Ourouadi, S. ; Hasib, A.; Moumene, H.; El Khiraoui, A.; Ouatmane, A.; Boulli, A.A. Bioactive Constituents and Antioxidant Activity of Moroccan garlic (Allium sativum L.). Journal of Natural Sciences Research. 2016, 6, 38-43.
(25) Chen, S.; Shen, X.; Cheng, S.; Li, P.; Du, J.; Chang, Y.; Meng, H. Evaluation of Garlic Cultivars for Polyphenolic content, and Antioxidant Properties. 2013, Plos One, 8, e79730. https://doi:10.1371/journal.pone.0079730
(26) Lu, X.; Ross, C.; Powers, J.R.; Aston, D.E.; Rasco, B.A. Determination of Total Phenolic Content and Antioxidant Activity of Garlic (Allium sativum) and Elephant Garlic (Allium ampeloprasum) by Attenuated Total Reflectance-Fourier Transformed Infrared Spectroscopy. Journal of Agricultural and Food Chemistry. 2011, 59, 5215-5221. https://doi.org/10.1021/jf201254f |

Round 2
Reviewer 1 Report
Author have added some details in the revised article and try to resolve the quarries. The article can be accepted for publication in the present form
Author Response
Thank you very much for your acceptance.
Reviewer 2 Report
The title should be revised, what does "and its potential uses as a postharvest agent" mean? postharvest agent doesn't make sense!
The proof reading and editing of the English language is required for all the text. There are many errors and mistakes in the text. Extensive editing of the English language and style is needed.
Section 2.5.1 should be added a reference for the method, Use this reference for the method: https://doi.org/10.1111/ijfs.15110
Figure 2,3, and 4: All letters should be superscript.
Author Response
Thank you very much for your comments to improve the quality of the manuscript. The required information is written in blue in the current manuscript version.
The title should be revised, what does "and its potential uses as a postharvest agent" mean? postharvest agent doesn't make sense!
- The title was changed due to the reviewer's suggestion and to reduce its word content.
The proof reading and editing of the English language is required for all the text. There are many errors and mistakes in the text. Extensive editing of the English language and style is needed.
- Though all manuscripts are perfectible, the new version was carefully checked in its English language and style. Please see the changes in blue in the current version.
Section 2.5.1 should be added a reference for the method, Use this reference for the method: https://doi.org/10.1111/ijfs.15110
- Reference was added.
Figure 2,3, and 4: All letters should be superscript.
- We guess that reviewer wants to write Tables instead of figures. Changes were made.

Round 3
Reviewer 2 Report
The revision has been carried out by authors completely.